# AIoT-Based Eyelash Extension Durability Evaluation Using LabVIEW Data Analysis

**DOI:** 10.3390/s25165057

**Published:** 2025-08-14

**Authors:** Sumei Chiang, Shao-Hsun Chang, Kai-Chao Yao, Po-Yu Kuo, Chien-Tai Hsu

**Affiliations:** 1Department of Electrical and Mechanical Technology, College of Technology, National Changhua University of Education, Bao-Shan Campus, No. 2, Shi-Da Rd., Changhua City 500208, Taiwan; chess1mail@gmail.com (S.-H.C.);; 2Department of Electronic Engineering, National Yunlin University of Science and Technology, Douliu 64002, Taiwan; kuopy@yuntech.edu.tw

**Keywords:** AIoT, eyelash extensions, LabVIEW, durability, cosmetic testing

## Abstract

This study introduces a novel platform, the Artificial Intelligence of Things Experimental Device Platform (AIoTEDP), to evaluate the durability of eyelash extensions under various environmental factors, including temperature, wind speed, and compression frequency. The experiment employs a three-factor full factorial design, utilizing LabVIEW to collect and analyze independent variables. The retention rate of eyelash extensions is the dependent variable for evaluating the durability. The proposed AIoTEDP regulates thermostats, stepper motors, and heating fans to simulate real-world eyelash extension usage conditions. Quantitative analyses are performed through visual assessments and image recognition technologies. The experimental results indicate that high temperatures and strong winds significantly reduce the durability of eyelash extensions. However, moderate bending damage (3000 repetitions) still allows for sufficient retention. This study validates the practicality and accuracy of the proposed AIoTEDP, showcasing its potential for innovative cosmetic testing systems to assess eyelash extension durability.

## 1. Introduction

With the steady growth of the global cosmetics market, consumer demands for the efficacy and quality of beauty products continue to rise. Among these, makeup durability has become one of the key indicators for evaluating product performance and influencing consumer repurchase decisions [1,2]. Eyelash extensions, also known as eyelash grafting, have become popular in the modern beauty industry. Users perceive eyelash extensions as a semi-permanent beauty enhancement that requires minimal effort to enhance natural aesthetics, making it increasingly favored, especially among young women. Since eyelash extensions effectively enhance eye contours and overall makeup appearance, they have become essential to many female consumers’ daily beauty routines [3]. However, after applying eyelash extensions, prolonged exposure to high temperatures and strong wind, frequent blinking by the user, and external factors such as rubbing, pressing, or bending can easily lead to issues such as shedding, warping, or detachment. These factors affect the actual effectiveness of the extensions.

Currently, durability assessments for eyelash extensions primarily rely on human trials or consumer surveys, which are highly subjective and lack consistency and reproducibility [4]. Researchers have recently introduced artificial intelligence (AI), image analysis, and machine learning technologies to detect makeup changes and extract skin image features [2]. However, dedicated testing systems specifically for evaluating the durability of eyelash extensions remain scarce. The Artificial Intelligence of Things (AIoT) integrates AI and Internet of Things (IoT) advantages, enabling real-time sensing and control of environmental conditions—such as temperature, wind speed, and physical forces—while transmitting and analyzing data. AIoT technology is widely applied across smart manufacturing, healthcare, and beauty technology, demonstrating high flexibility and precision [2]. In response to the new technology development, this study tries to incorporate AIoT concepts and develop an intelligent simulation system that evaluates eyelash extensions’ durability under various environmental conditions. This study fills a scholarly gap in cosmetology regarding the physical properties of eyelash extensions. It establishes a makeup durability assessment testing platform for eyelash extensions that has not yet been developed within the Taiwanese beauty industry.

Based on the above motivations and understanding, this study has three objectives, as follows: 1. to assess the feasibility of this proposed AIoT experimental device platform (AIoTEDP) and standardize the durability testing for eyelash extensions; 2. to evaluate the effects of these conditions on eyelash extension durability through a three-factor full factorial experimental design; and 3. to establish a standardized, reproducible, and commercially viable intelligent testing platform for eyelash extension durability.

## 2. Literature Review

### 2.1. Development Trends in Eyelash Extension Makeup Durability Assessment Methods

Eyelash extension products have been gaining increasing attention in the beauty market because they effectively enhance eye definition and add depth to the overall makeup look [4]. However, the durability of eyelash extensions is highly susceptible to external environmental factors, such as temperature fluctuations, wind friction, frequent blinking, and actions like rubbing, pressing, or bending. These influences can lead to lifting, deformation, or detachment, ultimately impacting the product’s overall performance and user satisfaction. Therefore, in recent years, establishing an objective, reliable, and reusable evaluation mechanism for eyelash extension durability has become an essential topic in beauty product testing. Traditional assessments of eyelash extension durability mostly rely on user feedback after real-world application or subjective evaluations by makeup artists based on the longevity of the look. While these methods offer practical reference value, variations among individuals and the uncertainty of environmental factors result in inconsistent and difficult-to-reproduce outcomes [4].

With advancements in technology, recent research has introduced scientific instrumentation for analysis. For instance, high-resolution digital imaging recognition has been used to track changes in eyelash extensions, curling angles, and detachment conditions. Additionally, residual eyelash extension volume changes have been employed to indicate durability [5]. AIoT technology has been incorporated to simulate environmental control and automate data recording, making evaluation processes more systematic and data-driven. These innovative approaches have introduced new methods for assessing eyelash extension makeup durability. However, the industry is still awaiting the establishment of a standardized testing mechanism, and no definitive conclusions have been reached to date.

### 2.2. Application of AI and Image Recognition Technology in Eyelash Extension Testing

With advancements in Artificial Intelligence (AI) and deep learning technologies, cosmetic testing research has increasingly adopted computer vision and image recognition techniques to enhance data collection accuracy and efficiency [6]. These technologies hold particular promise in eyelash extensions, where post-application changes—such as subtle deformations, curling angles, or localized detachment—are challenging to detect with the human eye. These minute details can be digitized by leveraging image recognition technology, allowing algorithms to conduct time-series analyses, thereby improving objectivity and reproducibility in evaluations.

Saiwaeo et al. (2023) [7] developed a Convolutional Neural Network (CNN) skin type classification model that utilizes Contrast-Limited Adaptive Histogram Equalization (CLAHE) to enhance image quality. By applying convolutional techniques for data augmentation, the model increases dataset diversity and achieves an accuracy rate of 95%. AIoT models can be further trained to identify varying degrees of eyelash extension adhesive degradation, providing real-time data feedback to cloud databases for dynamic trend analysis and predictive modeling [8,9]. These applications contribute to the standardization and automation of eyelash extension durability assessments, reducing ethical concerns and time costs associated with human trials and aligning with the development trends in smart manufacturing and beauty technology.

This study integrates the aforementioned CNN technology with Artificial Intelligence of Things (AIoT) to develop an innovative experimental device that leverages image recognition for automated eyelash extension durability assessment. This approach enhances testing efficiency and addresses the subjectivity inherent in traditional human evaluations.

### 2.3. Development of AIoT Applications in Eyelash Extension Durability Assessment

Artificial Intelligence of Things (AIoT) integrates AI and Internet of Things (IoT) infrastructure to enhance data management, analysis, and human–machine interaction, increasing efficiency in various applications. AIoT has become a crucial technology in intelligent cosmetic testing and quality control. Compared to traditional durability assessment methods that rely on manual observation or static instrument measurements, AIoT offers real-time data collection, automated control, and remote monitoring capabilities. It simultaneously enables environmental simulation and image recognition, providing significant advantages in eyelash extension durability testing [8].

The retention rate of eyelash extensions refers to the proportion of false lashes that remain securely attached to natural lashes after multiple compressions. This metric is used to evaluate eyelash extension durability. Environmental variables and user habits highly influence retention rates. For instance, high temperatures can soften the adhesive, strong winds can alter the curvature of extensions, and repetitive blinking or friction creates mechanical stress that may lead to breakage or detachment. Under these complex conditions, manual testing methods struggle to provide accurate and reliable durability data for measuring eyelash extensions.

This study builds the AIoTEDP module by referencing the literature discussed above. By integrating AIoT technology with heat modules, wind speed controllers, stepper motor pressure simulators, and temperature sensors, the researchers developed an AIoT experimental device platform (AIoTEDP) that realistically simulates environmental conditions affecting eyelash extensions. The proposed AIoTEDP integrates real-time image capture and AI-based image recognition modules to automatically analyze eyelash detachment, curvature deformation, and makeup coverage changes. It uploads analytical results to a cloud server for time-series data storage and analysis, ensuring the accuracy of experimental results. Additionally, the AIoTEDP enables dynamic reproduction of real-world usage scenarios when needed for research.

## 3. Materials and Methods

This study employs an experimental research approach to investigate the makeup durability of eyelash extensions under various environmental conditions. By utilizing a novel AIoTEDP developed by the researchers, the study aims to establish a quantifiable and reproducible testing platform for assessing eyelash extension durability. The research design focuses on three key variables: 1. temperature, 2. wind speed, and 3. compression frequency (to simulate blinking and mechanical friction). This study uses a three-factor full factorial experimental design to evaluate these variables’ impact on eyelash extension’s durability.

### 3.1. Selection of Experimental Variable Parameters

The experimental setting for this study is located in Taiwan, an island in the western Pacific rim. Given Taiwan’s island geography and its position within a monsoon climate zone and influences from mountainous and marine climates, annual temperatures range from approximately 0 °C to 40 °C [10]. Therefore, based on historical average temperatures published by the Central Weather Bureau of Taiwan, this study selects temperature levels commonly encountered by users in daily environments, setting the temperature variable at 15 °C, 25 °C, and 35 °C. Additionally, the typical wind speed experienced by eyelash extension users in outdoor settings ranges from 3.5 m/s to 24.4 m/s [11]. This study references the maximum wind speeds recorded for typhoons in this island by the Central Weather Bureau of Taiwan to establish reliable experimental conditions. Therefore, we set the wind speed parameters at 3.4–5.4 m/s, 10.8–13.8 m/s, and 20.8–24.4 m/s in the experimental design.

Furthermore, through a field survey conducted with 30 experienced eyelash technicians, followed by a formal review by five academic experts, this study determined, with 95% consensus, the appropriate levels for compression-induced stress simulations. The selected compression frequency levels are 1000 times, 3000 times, and 5000 times, reflecting repeated eyelid movements such as blinking and friction. Based on these parameters, this study establishes a three-factor, three-level experimental design to systematically assess the impact of environmental conditions on eyelash extension durability.

### 3.2. CNN Model Architecture and Evaluation

To evaluate the durability of eyelash extensions under various environmental conditions, this study developed an integrated experimental framework that combines simulation testing with AI-based image analysis. The system uses a simulated eyelash extension module to replicate real-world application scenarios, focusing on evaluating eyelash retention, curvature maintenance, and overall makeup integrity. Furthermore, it is embedded in a LabVIEW interface, enabling instant visual recognition and quantitative analysis of eyelash extension durability. This study incorporates a Convolutional Neural Network (CNN)-based image classification model into the AIoTEDP to enhance the objectivity and accuracy of eyelash extension durability assessment [12,13]. We trained the CNN to detect and classify eyelash extension conditions from captured images under varying environmental and stress conditions. The model aimed to differentiate between three eyelash conditions: (1) stable adhesion, (2) partial detachment, and (3) complete detachment.

Model Architecture

The CNN model consists of the following layers [14,15]: (1) Input Layer: 128 × 128 grayscale images, preprocessed from raw image captures; (2) Convolutional Layer 1: 32 filters, 3 × 3 kernel, ReLU activation, followed by 2 × 2 max-pooling; (3) Convolutional Layer 2: 64 filters, 3 × 3 kernel, ReLU activation, followed by 2 × 2 max-pooling; (4) Convolutional Layer 3: 128 filters, 3 × 3 kernel, ReLU activation, followed by 2 × 2 max-pooling; (5) Fully Connected Layer 1: 128 neurons, ReLU activation; and (6) Fully Connected Layer 2: 3 neurons, Softmax output. We selected this architecture for its simplicity and performance balance on small datasets. The convolutional layers in the proposed CNN architecture are designed to progressively learn spatial hierarchies of eyelash extension features. The first convolutional layer primarily extracts low-level features such as edges, contours, and texture patterns from the lash and adhesive boundary regions. The second convolutional layer captures more complex patterns, including localized detachment gaps, curvature variations, and small adhesive degradation marks. The third convolutional layer integrates these mid-level features to identify higher-order structural characteristics, enabling the model to distinguish between stable adhesion, partial detachment, and complete detachment with improved accuracy.

2.Dataset and Labeling

The image dataset was constructed from 1200 labeled images captured during a test run of the AIoTEDP: (1) 400 images labeled as “Stable adhesion”; (2) 400 images labeled as “Partial detachment”; and (3) 400 images labeled as “Complete detachment”. All images were annotated manually by a team of three certified eyelash experts to verify the labeling accuracy. The dataset was balanced across all classes. To ensure dataset representativeness, the random selection process for training (70%), validation (15%), and testing (15%) sets was stratified by class. This preserved the original class balance of 400 images per category across all subsets, ensuring that each set contained an equal proportion of “Stable adhesion,” “Partial detachment,” and “Complete detachment” samples.

3.Training, Validation, and Testing

The dataset was randomly divided into a (1) Training set: 70% (840 images); (2) Validation set: 15% (180 images); and (3) Testing set: 15% (180 images). The model was trained using the Adam optimizer with a learning rate of 0.001, using categorical cross-entropy loss for 50 epochs and a batch size of 32. Early stopping was applied to prevent overfitting.

4.Performance Evaluation

The performance evaluation methods used in this study included (1) Validation accuracy testing; (2) Testing accuracy verification; and (3) Precision/Recall/F1-score assessment. These methods are designed to reduce potential visual errors in human inspection, particularly in identifying “partial detachment” and “complete detachment.” The system effectively classifies eyelash extension detachment status by employing a CNN model for the above evaluation tasks and provides reliable data for subsequent statistical analysis in LabVIEW.

### 3.3. Establishment of Experimental Apparatus and System Architecture

#### 3.3.1. Experimental System Design

LabVIEW, developed by National Instruments, is a graphical programming language widely applied in data acquisition (DAQ), experimental control, and real-time data visualization [16,17], making it a valuable tool in industrial automation, medical devices, and scientific research [18]. Given its ability to rapidly develop stable and efficient systems, LabVIEW plays a crucial role in this study, supporting three core functionalities [19,20]:Integration of the data acquisition system (DAQ):(1) Utilizing NI DAQ and Arduino, capturing temperature, wind speed, and step-motor signals and recording; (2) LabVIEW functions as the intermediary software, performing real-time data visualization and storage, while transmitting collected data to a cloud-based analytical platform.Experimental parameter configuration and control [11]: The LabVIEW user interface allows for parameter configuration, including (1) Temperature: 15 °C, 25 °C, 35 °C; (2) Compression frequency: 1000, 3000, 5000 repetitions; (3) Wind speed: 3.4–5.4 m/s, 10.8–13.8 m/s, 20.8–24.4 m/s; (4) LabVIEW transmits signals to the Arduino master controller, which drives stepper motors and fan modules accordingly.Image capture and automated control: (1) Every 10 min, the camera module captures images of the eyelash extension module; (2) Captured images are timestamped and synchronized with experimental conditions, serving as AI-based image analysis data for subsequent comparisons.

#### 3.3.2. Experimental System Components

The AIoT experimental device platform (AIoTEDP) is designed to achieve the research objectives. The system consists of the following components: (1) Control unit: Arduino control board; (2) Stepper: Motor-driven compression module; (3) Temperature and wind speed simulation: Heating and fan module (heating carpet, thermistor, anemometer); (4) AIoT sensor feedback system: Image capture and cloud-based analysis; (5) Human–machine interface: LCD and control knob; and (6) Graphical identification: Real-time image capture and AI recognition unit.

This study references circuit designs from Kuo et al. (2023) [21] and Kuo et al. (2024) [22] to adapt the experimental modules to specific research needs. The configuration is illustrated in Figure 1, which presents the electrical signal detection device for evaluating eyelash extension compression damage retention rate. Figure 1 highlights the sensor signal amplification and data acquisition module, which functions as follows: (1) It measures weak signals generated by sensors within the eyelash extension simulation module; (2) Signals are amplified and processed using a common emitter amplifier and readout circuit; (3) The DAQ module acquires the signal and transmits it to LabVIEW for further analysis.

In Figure 1, the TR is a typical N-P-N switching transistor (2N2222), while V_R_ is a synchronized displacement resistor (V_R_ = 10 KΩ) installed alongside the eyelash extension module. When the stepper motor compresses the eyelash extension module in a circular motion, V_R_ fluctuates between 10 KΩ and 0 Ω. R_X_ and R_B_ are both 1 KΩ resistors. When V_R_ = 0 Ω, TR is active, and V_01_ = 5 V (logical state = 1). When V_R_ = 10 KΩ, TR is inactive, and V_01_ = 0 V (logical state = 0). The V_01_ output signal serves as the input for the readout circuit, facilitating accurate signal detection and processing for further analysis.

This study references sensor schematic designs from Kuo et al. (2024) [23] and Chou et al. (2024) [24], adapting them to fit the operational requirements of various experimental modules. Figure 2 illustrates the sensor signal acquisition and environmental control logic in the AIoT experimental device platform (AIoTEDP), detailing how sensor data from the simulated eyelash device is amplified by an instrumentation amplifier and transmitted to the LabVIEW software platform for processing. This system enables environmental parameter simulations through a control loop and performs mechanical operations via a stepper motor, simulating compression testing on eyelash extensions.

The experimental setup functions as a closed-loop feedback control system, where the main components and their functionalities include the following:Simulated eyelash device: The core module of the system, designed to mimic the physical response of eyelash extensions in real-world applications, including (1) a sensor unit that detects electrochemical signal output (V_o1_) related to eyelash compression deformation as shown in Figure 2; (2) a readout circuit, consisting of (a) three operational amplifiers LM741 (A_1_, A_2_, A_3_) forming an instrumentation amplifier that amplifies the weak voltage differential from the sensor (V_o1_) and (b) resistors (R_1_, R_2_, R_3_) incorporated to set gain parameters for signal amplification; (3) a data acquisition module (DAQ) that (a) converts the amplified signal (V_o_) into digital data and transmits it to a computer running LabVIEW software for processing and (b) simultaneously feeds acquired data into control components for system adjustments; (4) a LabVIEW system DAC: the core control software of the experiment, responsible for data reception, image synchronization, experimental parameter adjustments, and environmental condition control (e.g., compression retention rate, temperature, and wind speed simulation); (5) a bending simulation control module that (a) uses a stepper motor to rotate and manipulate eyelash extensions, replicating mechanical friction from blinking motions and (b) includes a heating fan and anemometer, which adjust temperature and wind speed, simulating various outdoor environmental conditions. The AIoTEDP modules and parameters are summarized as follows:Wind speed simulation module (fan + anemometer + voltage control)

Using PWM control and anemometer-based testing, the fan speed is adjusted to maintain preset wind speed levels at 3.4–5.4 m/s, 10.8–13.8 m/s, and 20.8–24.4 m/s, ensuring consistent wind conditions during the experiment.

3.Temperature simulation module (heating carpet + thermistor + thermostat)

The heating module is set to three different temperature levels: 15 °C, 25 °C, and 35 °C. The thermistor and thermostat provide real-time feedback, ensuring that the experimental environment maintains the preset temperature conditions.

4.AIoT sensor feedback system (image capture + cloud analysis)

A high-resolution USB camera captures images at scheduled intervals throughout the test, analyzing changes in eyelash extension durability via computational processing. The results are uploaded to a cloud platform for statistical analysis and record keeping.

5.Human–machine interface (LCD + control knob)

It provides researchers with an intuitive user interface to adjust environmental parameters and monitor experimental progress, improving operational efficiency.

#### 3.3.3. Layout of Experimental Apparatus and Experimental Materials

This study utilizes a custom-built AIoT automatic control system to regulate a thermostatic chamber, stepper motor, and heating fan, simulating real-world usage conditions. Through visual analysis and image recognition technology, quantitative assessments are conducted. The entire AIoT experimental device platform (AIoTEDP) includes components such as the Arduino control unit board, stepper motor module, simulated eyelash structure, and control circuit-related modules, as displayed in Figure 3.

As shown in Figure 3, the AIoTEDP corresponds to the following equipment and materials:(a)Stepper motor assembly. (Tricore Corporation, Changhua, Taiwan)

This study utilizes a self-developed eyelash manipulation experimental module to conduct various simulation conditions defined in the research design. The module consists of a stepper motor, an eyelash manipulation film, and an eyelash breakage detection sensor. It records and analyzes the number of manipulations before the eyelashes break and fall off, providing data to LabVIEW for further analysis and application.

(b)Eyelash extension device module and materials. (Cyber Lashes, Taipei, Taiwan)

In Figure 3a, the researchers developed the eyelash extension device module. We purchased the simulated material for bonding false eyelashes with natural eyelashes from a beauty materials company, Cyber Lashes in Taipei, Taiwan, featuring animal-derived mink hair and lightweight flat lashes (specifications: diameter 0.15 cm) with Suzy Power black adhesive (KS Bond in Taipei, Taiwan). Due to limitations in real human eyelash sampling, mink hair was used as a substitute for human eyelashes in the experiment design. Mink eyelashes were bonded with lightweight flat lashes using Suzy Power black adhesive, creating 200 fiber-based eyelash extensions for experimental testing, forming the eyelash extension module used in this study. The detailed material specifications of the eyelash extension device module in Figure 3a are: (1) Animal-based mink fur (substitute for human eyelashes): 3D layered mink fur eyelashes, Mink 61, purchased from Shopee, Taiwan; (2) Lightweight flat sashes (Extended false eyelashes): Synthetic eyelashes, flat lashes, lightweight, single lash (Specifications: Diameter 0.15 cm), purchased from Shopee, Taiwan; 3) V’dore black adhesive H_06_: Drying speed of 0.5–1 sec; relatively thin consistency; purchased from Shopee, Taiwan (Cyber Lashes, Taipei, Taiwan).

(c)Thermostatic device and temperature controller. (Thermoway Industrial Co., Ltd.)

The temperature module comprises a heating carpet, a thermistor, and a thermostat. The temperature control and measurement range of these three components is −10 °C to 50 °C (Measurement error: ±2 °C).

(d)Heating carpet. (Formosa FCFC Carpet Corporation, Changhua, Taiwan)

Providing the needed temperature that the experimental conditions required.

(e)Microcomputer (with LabVIEW) and sensor interface.

The researchers developed the AIoT sensor feedback system and interfaced it with the installed LabVIEW 2018 software to a microcomputer (ASUS VivoBook, Taipei, Taiwan). This module provides image capture to monitor compression-induced retention changes in eyelash extensions. The system integrates a user interface, LCD, and control unit, enabling cloud-based analysis.

(f)Adjustable electric hair dryer. (Shengyi Technology Co., Ltd., Taipei, Taiwan)

Accompanied by the heating carpet, it provided the temperature that the experimental conditions required.

(g)Anemometer (AZ Instrument Co., Ltd., Taichung, Taiwan) and Apple iPhone 16.

The wind speed control module consists of a fan and an anemometer. Figure 3e is a variable-speed hairdryer, configurable according to the study’s wind speed requirements (three speed settings; maximum wind speed: 67 m/s). The anemometer has a wind speed measurement range of 0.3–30 m/s (measurement error: ±5%). The Apple iPhone 16 is a synchronized data acquisition device for AIoTEDP information transmission.

### 3.4. Experimental Methods and Procedures

LabVIEW captures data based on this experiment’s three independent variables (temperature, wind speed, and compression frequency). Using a designed image evaluation model, the system calculates the average retention rate percentage of eyelash extensions, the dependent variable for durability assessment. The collected data is processed using SPSS 22.0 to analyze variance (ANOVA) to examine potential interactions between the variables, determining how temperature, wind speed, and compression frequency influence the durability of eyelash extension.

This study adopts a three-factor full factorial experimental design to manipulate three independent variables: temperature, wind speed, and compression frequency. Each factor is assigned three levels. With a 3 × 3 × 3 factorial combination, 27 distinct experimental conditions are generated. For each condition in the study design, the AIoTEDP is used for experimental manipulation. The experimental procedures are explained as follows:

First, before testing, eyelash extension products are pre-attached to the simulated eyelash module. Once temperature and wind speed parameters are set, the system automatically controls the heating and fan modules to create the designated environmental conditions. Second, the stepper motor initiates the specified compression cycles, replicating the mechanical stress caused by blinking and friction in daily use. Each condition is repeated three times to minimize operational errors, ensuring data stability and reliability. Third, during the simulation process, the AIoTEDP utilizes a built-in high-resolution camera module, which captures images of the simulated eyelash extension module every ten minutes. Fourth, the collected data is then transmitted to an image analysis system for quantitative assessment of eyelash extension durability. Fifth is the eyelash extension duration evaluation; the evaluation parameters include: (1) retention area percentage, (2) curing angle variations, and (3) signs of adhesive degradation. All experimental data is synchronized and uploaded to a cloud platform, where it is organized and statistically analyzed. The retention rate of eyelash extensions, obtained from image analysis, is the primary dependent variable for assessing extension durability.

To further examine the effects of temperature, wind speed, and compression frequency, this study applies analysis of variance (ANOVA) to determine possible interactions between the three independent variables. Through result interpretation and discussion, this study identifies the least favorable and most favorable environmental condition combinations for eyelash extension durability. This experimental design replicates real-world application scenarios and enhances testing efficiency and accuracy through the AIoTEDP. Fortunately, at the end of the experiment, this system offers an innovative evaluation tool for the beauty industry, enabling standardized assessments of eyelash extension durability.

## 4. Results and Discussion

### 4.1. Three-Factor Full Factorial Experimental Analysis

This study employs a three-factor full factorial experimental design to evaluate the impact of three independent variables (temperature, wind speed, and compression frequency) on the dependent variable (eyelash extension durability). The experiment consists of 27 trials, systematically combining all three independent variables in different conditions, with each trial repeated three times. The image evaluation model generates the average retention rate percentage for eyelash extensions, serving as the basis for determining durability.

The independent variables are set as follows: 1. Temperature (3 levels): 15 °C (low), 25 °C (normal), 35 °C (high); 2. Wind speed (3 levels): 3.4~5.4 m/s, 10.8~13.8 m/s, 20.8~24.4 m/s; 3. Compression frequency (3 levels): 1000 times, 3000 times, 5000 times. After completing the experiments, 3 × 3 × 3 = 27 experimental datasets were obtained, as summarized in Table 1.

#### 4.1.1. Temperature Effect Analysis and Discussion

The data in Table 1 shows that an increase in temperature reduces the average retention rate of eyelash extensions. Under constant compression frequency and wind speed, durability declines by approximately 10% as the temperature rises from 15 °C to 35 °C. For instance, at 1000 compression cycles and 3.5~5.4 m/s wind speed, the average retention rate drops from 92.5% (15 °C) to 85.3% (35 °C).

#### 4.1.2. Analysis and Discussion of Wind Speed Effects

From Table 1, we can observe that wind speed significantly impacts the retention rate of eyelash extensions. Under conditions where temperature and the number of bending cycles are controlled, an increase in wind speed results in decreased durability of the eyelash extension adhesive. For instance, at 15 °C and 1000 bending cycles, when wind speed increases from 3.4–5.4 m/s to 20.8–24.4 m/s, the retention rate drops from approximately 92.5% to 88.0%. This trend remains consistent across other temperature and bending conditions, indicating that higher wind speeds make the adhesive more susceptible to external interference, leading to detachment or warping.

The findings of this study align with the principles of physics and mechanics. The increase in wind speed induces shear stress and dynamic airflow disturbances [25], which amplify the friction and oscillation frequency at the adhesive junction between false eyelashes and natural lashes. Consequently, these effects contribute to adhesive surface degradation and polymer fatigue.

#### 4.1.3. Analysis and Discussion of Compression Frequency

Table 1 shows that the average retention rate of eyelash extensions at 3000 compression cycles is higher than at 5000 cycles and even close to or slightly higher than at 1000 cycles (as seen in trials 4 and 10). This finding suggests that moderate compression might enhance eyelash extension durability, while excessive compression still leads to damage at the adhesive junction.

#### 4.1.4. Analysis of Temperature, Wind Speed, and Compression Frequency Trends on Eyelash Extension Retention Rate

This study simultaneously considers the effects of temperature, wind speed, and compression frequency to analyze their combined impact on eyelash extension retention rate (makeup durability). The findings, illustrated in Figure 4, reveal a declining trend in retention rate under high temperature and high wind speed conditions.

As shown in Figure 4, increasing temperature and wind speed decrease the eyelash extension retention rate. This finding aligns with established material science principles, which suggest the following: (1) High temperatures soften eyelash extension adhesive, reducing its effectiveness. (2) Strong wind introduces mechanical stress, weakening the bond between false and natural lashes. (3) The combined impact of heat and wind further deteriorates adhesive integrity.

Regarding compression frequency, the study finds that when the compression cycle is set to 3000 repetitions, the retention rate is higher than at both 1000 and 5000 cycles. Researchers hypothesize that specific mechanisms may be at play when compression cycles are set to 3000, which requires further investigation. Possible explanations for the nonlinear retention phenomenon in eyelash extension adhesive are as follows:Potential redistribution of adhesive: Mechanical stress may promote a more uniform distribution of the adhesive layer, enhancing bonding strength between false and natural eyelashes [26,27,28].Possible material activation: The eyelash extension adhesive may exhibit thixotropic properties—the compression-induced deformation from the stepper motor may temporarily reduce viscosity, improving wetting and adhesion [29,30].Possible interface adaptation: Researchers speculate that repeated 3000-cycle compression may optimize the contact area between the adhesive and the eyelash surface, compensating for initial weak bonding points [28,31].

This experiment revealed a nonlinear phenomenon when compression cycles were set at 3000, contradicting conventional material science principles. Therefore, researchers also hypothesize that this unexpected result may be related to the chemical properties of the selected adhesive. Future research should test multiple eyelash extension adhesives and conduct a broader, comprehensive analysis to clarify the underlying mechanisms.

Based on the data in Table 1 and Figure 4, the following explanations can be derived:Under a fixed compression frequency, an increase in temperature significantly reduces eyelash extension retention rate.Under a fixed temperature condition, increasing compression cycles from 1000 to 3000 results in minimal change in the retention rate, but beyond 5000 cycles, the retention rate drops noticeably.The optimal retention rate is observed within the temperature range of 15 °C to 25 °C, with compression cycles centered around 3000 repetitions.The worst combination for retention rate occurs under 35 °C temperature and 5000 compression cycles, coupled with high wind speed conditions.

This study finds that a moderate compression frequency (3000 cycles) combined with a low-temperature environment leads to the best eyelash extension retention rate, ensuring maximum durability. However, excessively high temperatures and increased compression cycles result in mechanical fatigue, reducing retention rate and ultimately decreasing durability, making makeup applications prone to failure.

### 4.2. LabVIEW Experimental Analysis

This study used a CNN-based image classification model to support objective detection of eyelash shedding status. During the experiment, the AIoTEDP platform automatically captured images every 10 min. These images were then classified into three categories: (1) stable adhesion, (2) partial shedding, and (3) complete shedding. The CNN model was trained on a dataset of 1200 labeled images, and performance evaluation was as follows: (1) validation set accuracy: 93.2%; (2) test set accuracy: 91.7%; and (3) precision/recall/F1 score (macro-average): 91.5%/91.8%/91.6%.

These results indicate that the CNN model can robustly classify the shedding status of eyelash extensions and provide reliable data for subsequent LabVIEW visualization and statistical analysis, demonstrating reliable classification performance under different environmental and stress conditions. The output of this model is used to calculate a time series retention rate for each trial, which serves as the dependent variable in subsequent statistical analyses, including analysis of variance. Replacing subjective visual inspection with AI-driven image analysis ensures greater consistency and reliability in durability assessments. These results indicate that the CNN model can robustly classify eyelash extension detachment status and provide reliable data for subsequent LabVIEW visualization and statistical analysis.

#### 4.2.1. Effects of Temperature and Wind Speed on Eyelash Extension Retention Rate Under Fixed Compression Frequency (1000 Cycles)

This study simulates compression frequency at 1000 cycles, examining how temperature and wind speed influence eyelash extension retention rate, represented by sensor output voltage waveforms. As illustrated in Figure 5, the trend curves correspond to three different temperature conditions (15 °C, 25 °C, and 35 °C) combined with three wind speed levels (3.5~5.4 m/s, 10.8~13.8 m/s, and 20.8~24.4 m/s). These curves reflect variations in retention rate, demonstrating how environmental factors impact eyelash extension durability. We processed the captured images through the CNN-based classification model. The model’s output classified eyelash status in each frame, enabling the calculation of a time-based stability index. The CNN model allowed faster identification of retention failure trends compared to manual inspection, enhancing the system’s real-time feedback capability.

Observations from the waveform in Figure 5 show the following findings:The variation in the average eyelash extension retention rate waveform is influenced by temperature.
(1)As temperature increases, the waveform exhibits an upward trend, suggesting that heat causes the adhesive layer to soften or enhances electrochemical activity, leading to a higher detected voltage.(2)This trend aligns with the findings in Table 1, where the retention rate significantly decreases at high temperatures (35 °C).(3)Higher temperatures may activate the adhesive, reducing overall bonding durability as environmental temperature rises.
2.The amplitude of the retention rate waveform is not correlated with wind speed.

As wind speed increases from 4 m/s to 22 m/s, the oscillation amplitude of the eyelash extension retention rate waveform decreases, becoming more stable. It suggests that stronger wind speed causes “dynamic interference” at the contact surface between false and natural eyelashes. Prolonged exposure to strong winds weakens the adhesive’s viscosity, leading to a decline in retention rate. This finding is consistent with Mu et al. (2023) [32], who studied the adhesion strength of ice layers on offshore wind turbine blades and found that higher wind speeds reduced the adhesion between the ice layer and the blade surface. This phenomenon may be attributed to increased shear forces, which disrupt the adhesion interface. A similar effect was observed in this study, where increased wind speed weakens the viscosity of eyelash extension adhesive. Additionally, this finding suggests that the amplitude of voltage waveforms does not correlate with the average retention rate of eyelash extensions.

#### 4.2.2. Effects of Temperature and Wind Speed on Eyelash Extension Retention Rate Under Fixed Compression Frequency (3000 Cycles)

This study utilizes sensor voltage waveforms captured by the LabVIEW-controlled AIoTEDP to analyze the impact of different temperature conditions (15 °C, 25 °C, 35 °C) and wind speeds (4 m/s, 12 m/s, 22 m/s) on sensor voltage outputs when compression cycles are set to 3000 repetitions. Figure 6 reveals that the voltage waveform amplitude decreases as temperature increases, demonstrating noticeable interference effects. This study observes a nonlinear relationship between sensor voltage variations (representing eyelash extension adhesive retention rate) and temperature and wind speed conditions. The findings, obtained using the LabVIEW-controlled AIoTEDP, suggest complex interactions between temperature, wind speed, and adhesive stability, which are discussed in further detail below.

The Relationship Between Voltage Waveforms, Temperature, and Wind Speed Is Nonlinear.
(1)Higher temperatures result in lower sensor voltage amplitude.As shown in Figure 6, when the ambient temperature increases from 15 °C to 35 °C, the peak-to-peak voltage amplitude of the sensor output significantly decreases. However, this trend is nonlinear—rather than a gradual decline, a sharp amplitude compression occurs at 35 °C, suggesting a potential adhesive failure in the eyelash extension bond.(2)The correlation between voltage amplitude and wind speed also exhibits a nonlinear relationship.The voltage waveform amplitude stabilizes at high wind speed (22 m/s), indicating reduced structural responsiveness to mechanical interference and deterioration of the adhesive mechanism. At low-to-moderate wind speeds, voltage amplitude fluctuations remain relatively smooth, suggesting that the adhesive maintains responsiveness to mid-range air pressure, exhibiting a threshold effect.(3)The correlation between eyelash retention rate and voltage waveforms is not a linear decline.Interestingly, under 3000 compression cycles, certain conditions (e.g., 15 °C + 4 m/s) result in a higher retention rate than 1000 compression cycles (87.6% vs. 92.5%), which contradicts conventional material science principles.Discussion on the Nonlinear Relationship Between Voltage Waveforms, Temperature, and Wind Speed.Based on detailed observations of experimental processes and variations, the researchers identified three possible causes for the nonlinear effects seen in voltage waveforms relative to temperature and wind speed:(1)Mechanical redistribution: A total of 3000 compression cycles generates moderate mechanical stimulation, leading to a more uniform distribution of the adhesive across the eyelash base. Increased bonding surface area enhances the retention rate. This reflects a “moderate stress activation” effect, stabilizing voltage waveforms while slightly increasing voltage amplitude.(2)Thermal softening-induced adhesive failure: At high temperatures (35 °C), the adhesive softens or exhibits enhanced thixotropic properties, losing its stress-response capability. This results in a significant decrease in waveform amplitude, resembling the rheological behaviors of non-Newtonian fluids.(3)Shear-rate dependent fatigue from wind speed interference: High wind speeds cause localized shear stress concentrations, accelerating adhesive fatigue. Voltage response waveforms stabilize, indicating diminished bonding integrity. This finding exhibits a nonlinear relationship, where wind speed does not directly correlate with adhesive deterioration but displays threshold-based failure characteristics.

This study finds that at 3000 compression cycles, sensor voltage waveform variations do not exhibit a linear decline but instead reveal a “plateau effect” or “reinforced viscosity phenomenon.” These intriguing findings suggest that mechanical pressure, environmental temperature, and wind speed interact nonlinearly. Future research should integrate materials science to explore the dynamic behavior of eyelash extension adhesives, incorporating microstructural analysis to validate nonlinear interaction mechanisms.

#### 4.2.3. Effects of Temperature and Wind Speed on Eyelash Extension Retention Rate Under Fixed Compression Frequency (5000 Cycles)

This study utilizes sensor voltage waveforms, captured by the LabVIEW-controlled AIoTEDP, to analyze the impact of different temperature conditions (15 °C, 25 °C, 35 °C) and wind speeds (4 m/s, 12 m/s, 22 m/s) on sensor voltage outputs when compression cycles are set to 5000 repetitions. Figure 7 illustrates the waveform trends under high compression fatigue conditions, showing how sensor voltage changes in response to environmental temperature and wind speed variations.

As environmental conditions worsen (high temperature and high wind speed), sensor voltage amplitude decreases significantly, and the waveform flattens.This finding indicates progressive adhesive failure, where the response capability of the eyelash extension adhesive to external interference is noticeably reduced. However, it might provide early warning indicators for adhesive deterioration, offering valuable insights into durability loss.

#### 4.2.4. Voltage Waveform Trends Under Fixed Temperature (25 °C) and Wind Speed (12 m/s) for Different Compression Cycles

Furthermore, this study simulates sensor voltage waveform variations under fixed environmental conditions (25 °C, 12 m/s) as compression frequency increases (1000, 3000, and 5000 cycles). Table 2 shows that (1) while the baseline voltage remains relatively stable, the peak-to-peak voltage amplitude decreases as compression cycles increase, and (2) this trend indicates that higher compression cycles significantly decrease the eyelash extension retention rate. At 1000 compression cycles, the waveform remains well-defined with strong amplitude, suggesting that eyelash extensions maintain structural stability under these conditions.

However, at 3000 compression cycles, the waveform amplitude slightly decreases, and its edges become less sharp, potentially indicating fatigue effects and microcrack formation in the adhesive layer. Finally, at 5000 compression cycles, the waveform flattens significantly, with amplitude shrinking to less than half of the lower compression cycle levels. This finding suggests a significant decline in adhesive bonding capability. The findings corroborate previous experimental results, demonstrating that the eyelash extension retention rate noticeably declines as environmental conditions become more extreme.

This study utilizes LabVIEW to analyze sensor output voltage waveforms. At the early stage of the experiment (1000 compression cycles), the adhesive between the false and natural eyelashes demonstrated strong adhesion and mechanical responsiveness. However, after reaching 3000 compression cycles, the waveform amplitude slightly declined. As compression cycles increased to 5000 repetitions, the waveform amplitude sharply dropped to only 40% of its initial level (at 1000 cycles). This indicates that the eyelash adhesive exhibits a degradation-based dynamic response under bending fatigue conditions, as shown in Figure 8. As the number of bending cycles increases, the peak-to-peak voltage amplitude progressively decreases, signifying the loss of adhesive retention and mechanical stability. The gel structure may have entered the post-fatigue zone at this stage, characterized by advanced material wear and deterioration.

This study concludes that a moderate compression frequency (3000 cycles) best preserves the integrity of voltage signals with minimal attenuation. The findings support that the adhesive interface may undergo activation or redistribution effects under moderate mechanical stress conditions. However, excessive mechanical cycling (5000 cycles) leads to a significant decline in signal amplitude, confirming the occurrence of mechanical fatigue and adhesive failure. Thus, voltage waveforms are a reliable quantitative indicator for simulating eyelash extension retention rates under normal usage conditions.

### 4.3. Analysis and Discussion of Results from the Three-Factor Full Factorial Experimental Design

This experiment manipulates nine condition combinations based on three independent variables (temperature, wind speed, and compression frequency). Using a designed image evaluation model, the system calculates the percentage of eyelash extension retention rate (dependent variable), serving as the basis for determining durability. All collected data undergoes analysis of variance (ANOVA) to examine the effects of each variable and their interactions on eyelash extension durability.

#### 4.3.1. Experimental Design and Research Hypotheses

The experimental design type is “3 × 3 × 3 factorial design.” The independent variables (X) are: Temperature: 15 °C, 25 °C, 35 °C; Wind speed: 3.4–5.4 m/s, 10.8–13.8 m/s, 20.8–24.4 m/s; and Compression cycles: 1000, 3000, 5000 repetitions. The dependent variable (Y) is the eyelash extension retention rate (%). The repetition arrangement is that each condition is tested thrice, totaling 81 experimental data points. The statistical hypotheses are as follows:

**H_0_:** No significant differences exist among factor levels (μ_1_ = μ_2_ = μ_3_).

**H_1_:** At least one factor level exhibits a significant difference.

#### 4.3.2. ANOVA Model Construction

This study adopts a three-factor mixed-effect model, represented by the following formula:(1)$$Y_{ijk}=\mu +a_{i}+\beta_{j}+\gamma_{k}+{\left(\alpha \beta \right)}_{ij}+{\left(\alpha \gamma \right)}_{ik}+{\left(\beta \gamma \right)}_{jk}+{\left(\alpha \beta \gamma \right)}_{ijk}+\varepsilon_{ijk}\;$$
where:

μ = Overall mean.

$a_{i}\;$= Main effect of temperature $\left(i\;=\;1,\;2,\;3\right)$.

$\gamma_{k}$ = Main effect of wind speed $\left(k\;=\;1,\;2,\;3\right)$.

Interaction terms = Includes all second-order and third-order interactions.

$\varepsilon_{ijk}$ = Random error (assumed i.i.d. N(0, $\sigma_{2}$)).

#### 4.3.3. Analysis of Variance (ANOVA)

This study utilizes the AIoT experimental device platform (AIoTEDP) proposed by this study to manipulate three independent variables—temperature, compression cycles, and wind speed. The study evaluates the durability of the dependent variable (eyelash extension retention rate) through a three-factor full factorial experimental design. A total of 27 experimental data points were collected. These were analyzed using ANOVA in SPSS 22.0 to examine the effects of the three independent variables on eyelash extension durability.

Table 3 presents the results of a three-factor ANOVA analysis, where temperature, wind speed, and compression cycles are treated as independent variables, and eyelash extension durability is the dependent variable. The findings are further explained and discussed as follows:

Each factor—temperature, wind speed, and compression cycles—exhibits an apparent main effect with a distinct trend. Temperature has the greatest influence, followed by compression cycles and wind speed.Interactions exist among the three factors: under the conditions of 35 °C, 5000 compression cycles, and 20.8~24.4 m/s wind speed, the eyelash extension retention rate reaches its lowest value (61.7%).The consistency across three repeated measurements is high, with minimal error margins, confirming the reliability of the data.

The statistical analysis using ANOVA reveals that the main effects of the three factors—temperature, wind speed, and compression cycles—are all highly significant (*p* < 0.01). These results indicate that each factor significantly impacts eyelash extension retention rate. Among them, temperature has the most pronounced effect (*F* = 35.21, *p* < 0.001), demonstrating that higher temperatures accelerate adhesive degradation, leading to a decline in retention rate. Additionally, this study finds that the interactions among temperature, wind speed, and compression cycles do not reach statistical significance. However, the Temp × Force interaction term approaches significance (*p* = 0.084), suggesting that a combined effect may still occur under specific temperature and compression cycle conditions. Future studies should expand the experimental sample size to validate this finding further.

The proposed AIoTEDP has successfully demonstrated its ability to accurately assess environmental and stress factors (temperature, wind speed, and compression cycles) influencing eyelash extension durability. With a strong scientific foundation, this AIoTEDP has the potential to be expanded into an innovative durability testing system for the beauty industry, contributing to the future development of eyelash extension evaluation technology.

#### 4.3.4. Experimental Findings and Discussion

This study employs a three-factor full factorial experimental design to systematically investigate the effects of temperature, wind speed, and compression cycles on eyelash extension durability. Additionally, by integrating AIoT technology, the researchers developed an intelligent detection platform, which was used for experimental analysis. The results indicate that all three independent variables significantly impact eyelash extension durability (dependent variable), each exhibiting an apparent main effect.

Independent Effects of Environmental and Control Variables.
(1)Temperature: High temperatures (35 °C) significantly reduce eyelash extension durability, especially when combined with high wind speed or frequent compression cycles. Under these extreme conditions, the lowest recorded retention rate was 61.7%. This finding aligns with previous research [33], which reports that eyelash extension adhesive softens at high temperatures, reducing adhesion strength.(2)Wind speed: Although a less dominant factor, strong winds (20.8~24.8 m/s) still contributed to a 3%–5% decrease in retention rate. This result suggests that outdoor exposure to wind-induced friction may cause a cumulative negative impact on eyelash extension durability; this finding is consistent with the results of Cheng et al. (2020) [11].(3)Compression cycles: A nonlinear effect was observed, marking an important discovery in this study.

Moderate compression (3000 cycles) resulted in a retention rate close to (or slightly higher than) the lower compression level (1000 cycles). At 15 °C, the retention rate at 3000 cycles was 87.6%, while at 1000 cycles, it was 92.5% [14,15]. We speculate that mechanical pressure from the apparatus during the experiment may have enhanced adhesive distribution, improving attachment strength. However, excessive compression (5000 cycles) led to structural fatigue, weakening the bond between false eyelashes and natural lashes.

2.Potential Effects of Interaction Terms.

The interaction among the three factors—temperature, wind speed, and compression cycles—did not reach statistical significance (*p* > 0.05). However, the interaction between temperature and compression cycles (*p* = 0.084) exhibits a marginal effect. For example, at 35 °C, increasing compression cycles from 3000 to 5000 resulted in a retention rate decrease of approximately 10%, greater than the decline observed at 15 °C (around 5%). This finding suggests that high temperatures may amplify the destructive effects of mechanical fatigue. Given this observation, future research should expand the sample sizes for further validation, exploring the rheological properties of eyelash extensions under extreme temperature and compression conditions.

3.Implications on the Value after the AIoTEDP is Verified

With the above findings and the discussion, the proposed AIoTEDP, in this study, integrates environmental and stress variables, conducting simulations and image analysis to overcome the subjectivity of traditional human assessments, and has been verified as an effective system. The system can precisely detect subtle signs of eyelash degradation through standardized controls and high-frequency image capture (every 10 min). Additionally, across three repeated experiments, data consistency remains high (error margin <1%), validating the reliability of this platform as a standardized eyelash extension durability testing tool. The research findings align with Barthe et al. (2021) [2], who highlighted the potential of AIoT in intelligent cosmetic testing, particularly in simulating complex environmental conditions for durability assessments.

This novel AIoTEDP clarifies the effects of environmental factors (temperature and wind speed) and stress variables on eyelash extension durability and validates the feasibility of AIoT technology in eyelash extension durability testing. By providing a scientifically grounded and efficiency-driven evaluation tool, this platform offers a promising foundation for the beauty industry, paving the way for the future implementation of an innovative durability testing system for eyelash extensions.

## 5. Conclusions and Future Research Recommendations

### 5.1. Conclusions

#### 5.1.1. Summary of Findings

This study provides specific quantitative data on the effects of temperature, wind speed, and compression cycles on eyelash extension durability. These findings can help beauty industry manufacturers develop longer-lasting eyelash extensions that are more suitable for general consumer use. Significant findings on environmental factors affecting eyelash extension durability are as follows:Temperature exhibits a negative correlation with eyelash extension durability:
(1)The experimental results indicate that temperature significantly impacts durability (*F* = 35.21, *p* < 0.001).(2)Under high-temperature conditions (35 °C), the retention rate declines noticeably, with more severe degradation occurring when combined with high wind speed and increased compression cycles.Wind speed also negatively affects eyelash extension durability:
(1)Although its effect is less significant than temperature and compression cycles, it is still statistically significant (*p* < 0.01).(2)Strong winds (20.8~24.4 m/s) accelerate eyelash extension detachment, especially in high-temperature environments.Moderate compression cycles result in the best retention rate:
(1)Moderate compression (3000 cycles) produces a retention rate similar to the lower compression level (1000 cycles).(2)However, excessive compression (5000 cycles) leads to structural damage, significantly reducing makeup durability.The interaction effects among the three factors are not statistically significant:

The interaction among temperature, compression cycles, and wind speed did not reach statistical significance. However, the interaction between temperature and compression cycles is close to significance (*p* = 0.084), suggesting a potential combined effect that warrants further investigation.

#### 5.1.2. Commercial Value of the AIoT Experimental Device Platform

Integrating a CNN image classification module was pivotal in enhancing the AIoTEDP platform. By accurately identifying the eyelash extension detachment status during testing, the CNN module enabled automated, real-time tracking of retention dynamics. The classification outputs were directly used to compute retention rates, reducing reliance on subjective visual inspection. Notably, the CNN model maintained over 91% accuracy on the test set and handled complex visual patterns consistently.

This study introduces a novel AIoTEDP, establishing a standardized testing method for eyelash extension durability with potential commercial applications. The platform can accurately simulate various environmental conditions and, through image recognition technology, quantify the durability of eyelash extensions over time. The experimental results show consistency and reproducibility, confirming its potential as a commercially viable standardized testing tool. Significant findings from the AIoTEDP are as follows:Compression cycles significantly influence the retention rate’s voltage waveform amplitude and durability.
(1)When eyelash extensions undergo 1000 compression cycles, the voltage oscillation waveform remains distinct and stable, indicating good adhesive performance.(2)As compression cycles increase to 3000 and 5000, the waveform gradually flattens and amplitude decreases significantly, demonstrating a damping effect.(3)This reflects the progressive deterioration of the adhesive’s electrical conductivity and structural integrity due to repetitive mechanical fatigue.
2.Retention rate voltage waveforms can be early warning indicators of eyelash extension failure.
(1)The study finds that when the waveform amplitude drops below 50% of the original peak-to-peak value, the corresponding retention rate falls below 80%.(2)This result indicates a strong correlation between retention rate waveform characteristics and adhesive condition.(3)This discovery provides a foundation for developing self-diagnosis digital detection modules, effectively monitoring equipment durability, and ensuring testing accuracy in future applications.

### 5.2. Research Limitations and Future Research Recommendations

While this experiment involved significant investments of resources, certain limitations remain due to material constraints. The nonlinear properties observed in this study warrant further investigation into the chemical composition of adhesives (e.g., cyanoacrylate-based formulas) and their rheological behavior under dynamic stress conditions. Below are some recommendations for future research:Expand the range of eyelash extension adhesives studied: Compare various commercial adhesive products’ physical and chemical properties to validate specific material responses.Enhance experimental conditions and increase sample diversity: (1) Introduce additional environmental factors such as humidity and ultraviolet exposure to simulate more complex real-world conditions. (2) Expand environmental parameters and evaluation criteria, including adhesive cracking patterns and warping angles.Apply the AIoTEDP to other cosmetic product testing: The proposed AIoTEDP can be a standardized durability testing system for various cosmetics, minimizing human subjectivity in evaluation. Integrating cloud-based data analytics, the AIoTEDP can build predictive models for makeup durability, helping manufacturers rapidly assess new product performance.

## Figures and Tables

**Figure 1 sensors-25-05057-f001:**
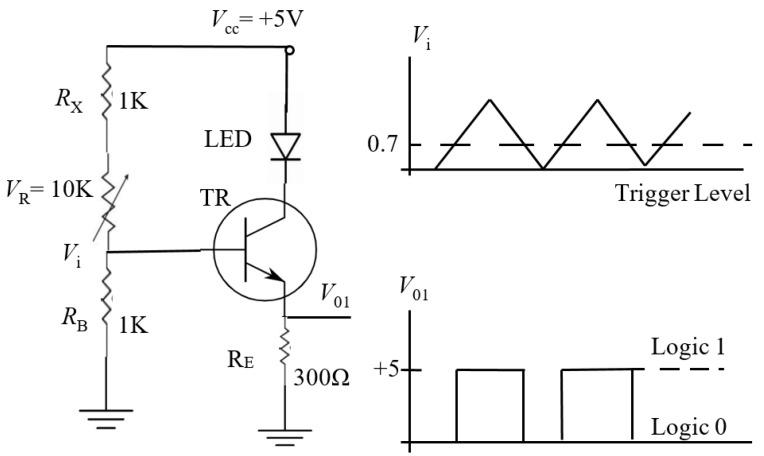
An electrical signal detection device assesses the eyelash extension compression damage retention rate.

**Figure 2 sensors-25-05057-f002:**
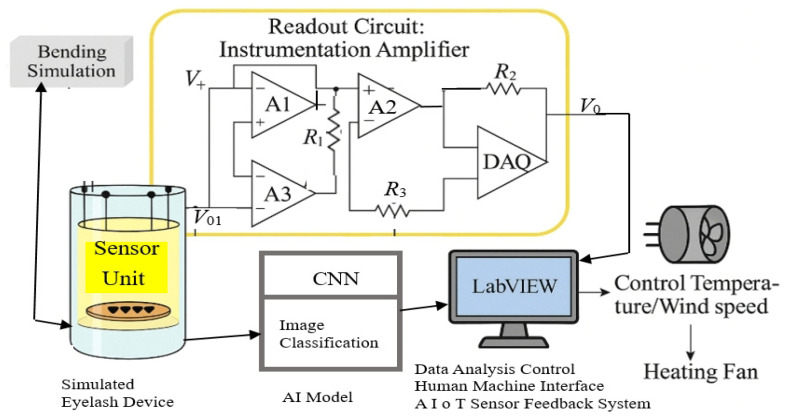
AIoT experimental device platform and signal readout circuit.

**Figure 3 sensors-25-05057-f003:**
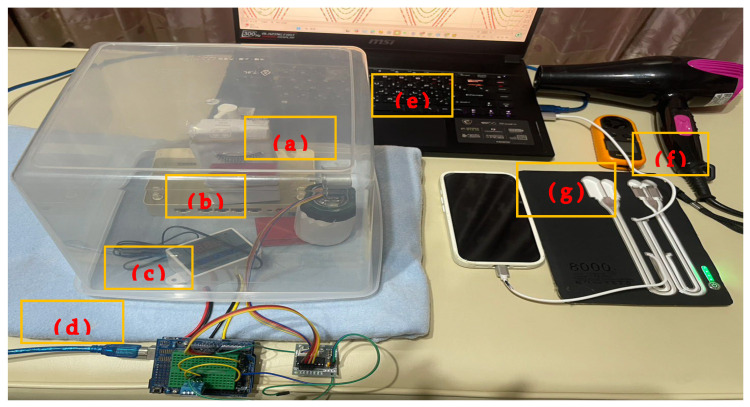
Layout of the AIoTEDP: (a) stepper motor assembly, (b) eyelash extension kit, (c) thermostatic device and temperature controller, (d) heating carpet, (e) microcomputer (with LabVIEW 2018 installed), (f) adjustable electric hair dryer, and (g) anemometer and Apple iPhone 16.

**Figure 4 sensors-25-05057-f004:**
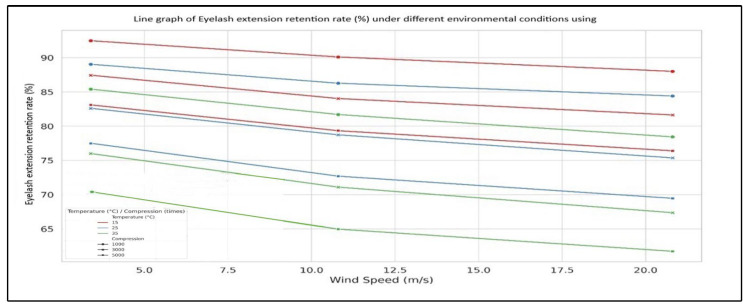
Line graph of eyelash extension retention rate under different environmental conditions using the LabVIEW-based AIoTEDP.

**Figure 5 sensors-25-05057-f005:**
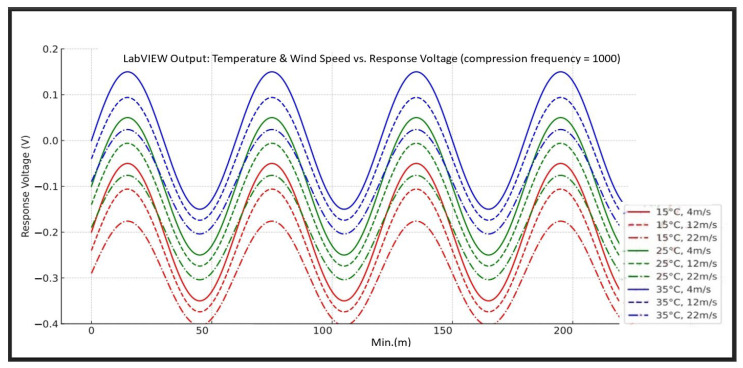
Effects of temperature and wind speed on eyelash extension retention rate under fixed compression frequency (1000 cycles).

**Figure 6 sensors-25-05057-f006:**
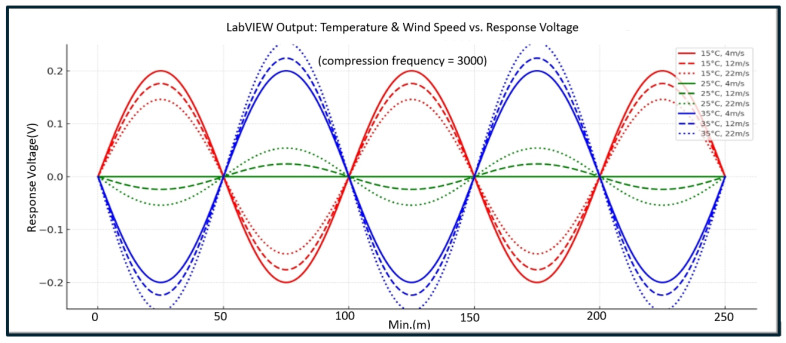
Effects of temperature and wind speed on eyelash extension retention rate under fixed compression frequency (3000 cycles).

**Figure 7 sensors-25-05057-f007:**
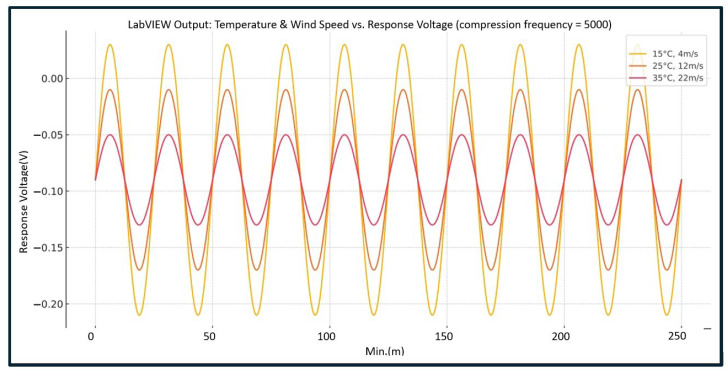
Effects of temperature and wind speed on eyelash extension retention rate under fixed compression frequency (5000 cycles).

**Figure 8 sensors-25-05057-f008:**
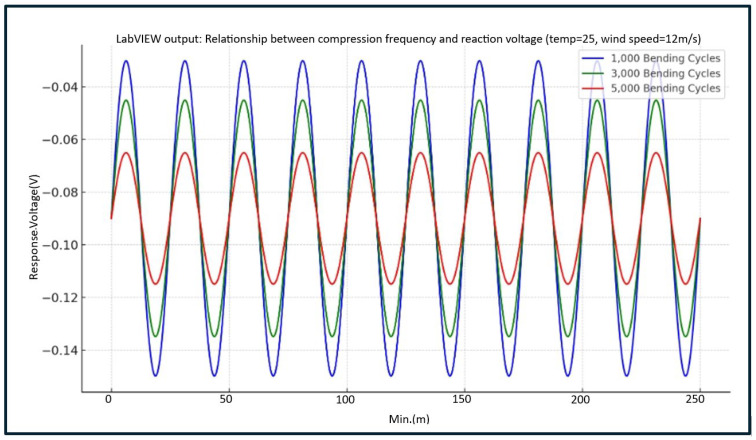
Sensor voltage waveform under fixed temperature (25 °C) and wind speed (12 m/s) for different compression cycles (1000/3000/5000 repetitions).

**Table 1 sensors-25-05057-t001:** Experimental data for the 27 trials are based on three independent variables.

Trial No.	Temperature (°C)	Compression Frequency	Wind Speed (m/s)	Retention Rate (%)		
				One Data	Two Data	Three Data
1	15	1000	3.4–5.4	92.5	92.4	92.5
2	15	1000	10.8–13.8	90.1	90.2	90.0
3	15	1000	20.8–24.4	88.0	87.9	88.1
4	15	3000	3.4–5.4	87.6	86.9	87.8
5	15	3000	10.8–13.8	84.3	83.9	83.9
6	15	3000	20.8–24.4	81.7	81.6	81.6
7	15	5000	3.4–5.4	83.2	83.3	82.8
8	15	5000	10.8–13.8	79.5	79.2	79.3
9	15	5000	20.8–24.4	76.8	75.9	76.5
10	25	1000	3.4–5.4	89.1	88.9	89.1
11	25	1000	10.8–13.8	86.4	86.3	86.1
12	25	1000	20.8–24.4	84.2	84.6	84.4
13	25	3000	3.4–5.4	82.7	82.6	82.5
14	25	3000	10.8–13.8	78.9	78.6	78.7
15	25	3000	20.8–24.4	75.3	75.4	75.4
16	25	5000	3.4–5.4	77.6	77.6	77.3
17	25	5000	10.8–13.8	72.8	72.8	72.5
18	25	5000	20.8–24.4	69.5	69.3	69.6
19	35	1000	3.4–5.4	85.3	85.4	85.5
20	35	1000	10.8–13.8	81.9	81.5	81.7
21	35	1000	20.8–24.4	78.5	78.4	78.4
22	35	3000	3.4–5.4	76.1	76.0	75.9
23	35	3000	10.8–13.8	71.2	71.1	71.0
24	35	3000	20.8–24.4	67.5	67.5	67.1
25	35	5000	3.4–5.4	70.4	70.4	70.2
26	35	5000	10.8–13.8	65.1	65.0	64.8
27	35	5000	20.8–24.4	61.8	61.7	61.7

**Table 2 sensors-25-05057-t002:** Comparison of eyelash extension retention rate based on sensor voltage waveform under different compression cycles (temp = 25 °C, wind = 12 m/s).

Compression Cycles	Baseline Voltage (Volts)	Peak-to-Peak Amplitude (Volts)	Waveform Durability	Interpretation
1000	Approx. −0.09	Approx. 0.12	Clear waveform, stable amplitude.	Strong adhesive durability.
3000	Approx. −0.09	Approx. 0.09	Waveform slightly flattens, edges become dull.	The adhesive begins to deteriorate.
5000	Approx. −0.09	Approx. 0.05	Waveform flattens, with almost no visible oscillation.	Adhesive strength significantly declines, nearing.

**Table 3 sensors-25-05057-t003:** Summary of the three-factor ANOVA results on eyelash extension durability.

Factors	Degrees of Freedom (*df*)	Sum of Squares (SS)	Mean Square (MS)	*F*-Value	*p*-Value
Temperature (Temp)	2	680.12	340.06	35.21	<0.001
Wind Speed (Wind)	2	148.67	74.34	8.56	0.002
Compression Cycles (Force)	2	212.45	106.23	11.78	<0.001
Temp × Force	4	87.24	21.81	2.26	0.084
Temp × Wind	4	64.12	16.03	1.86	0.133
Force × Wind	4	59.38	14.85	1.65	0.162
Temp × Force × Wind	8	45.71	5.71	0.89	0.527
Error	54	520.18	9.63	—	—
Total	80	1817.87	—	—	—

## Data Availability

The data supporting this study’s findings are available on request from the corresponding authors. The data are not publicly available due to privacy restrictions.

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
