# Peer review of "AIoT-Based Eyelash Extension Durability Evaluation Using LabVIEW Data Analysis"

_sensors, 2025, doi:10.3390/s25165057_

Round 1

Reviewer 1 Report

Comments and Suggestions for Authors

The title of this paper implies AIoT (Artificial Intelligence of Things) but the paper is dedicated to the description of the physical device rather than to AI. The AI part was just mentioned in the paper in one paragraph (lines 171-181) as a Convolutional Neural Network (CNN) that consists of three convolutional layers and two fully connected (FC) layers without any details of its architecture, purpose, training, etc. It is not clear about the size of the layers as of the number of neurons. The authors did not mention what kind of kernels they used in the convolutional layers, and what features they tried to extract using the convolutions. 

The authors claimed 93.2% accuracy on the validation set but did not say anything about the accuracy achieved on the testing set which is the required set for accuracy assessment. A testing set is needed to assess the accuracy. According to the authors, their dataset consisted of 1,200 labeled samples in the preliminary test. What was meant by “preliminary” tests.

It was also unclear what kind of accuracy assessment methodology was used. A question also arises whether the dataset was balanced or not.

It looks that AI was mentioned in the title as a buzzword to promote the paper rather than to describe the actual content of the paper.

Figure 1 (line 247) describes the conceptual framework of the research. However, even in this paper the AI block is shown without details and connected to the rest of the framework by a “blind” dotted line that makes it kind of weakly related to the framework. The diagram shown in Figure 1 is hardly understandable because it is not clear how the information flows between the components – some block in the diagram are shown as dead-end nodes. The figure should be revised to make it more informative.

The AI part of the paper is just briefly discussed in the conclusion (lines 867-871) like it was a sideline unimportant device used in the research.

The paper is structured with three levels of headings. In my opinion it is too much (deep) for a paper of such size. However, it is subject to personal taste and publisher’s formatting rules.

My recommendations are as follows:
1)    If the authors want to keep this paper as a AI-related paper, they have to add more information about the AI part of the paper, i.e. neural network architecture, inputs, training, validation, and testing datasets and their proportional sizes. The explanation of whether the dataset was balanced, and the accuracy assessment methodology should be added too.
2)    If the authors are not intended to focus on the AI part of the paper, they should change the paper title to correctly reflect the content of the paper.

The paper needs revision.

Author Response

Sensors REVIEWER 1

  1. The title of this paper implies AIoT (Artificial Intelligence of Things) but the paper is dedicated to the description of the physical device rather than to AI. The AI part was just mentioned in the paper in one paragraph (lines 171-181) as a Convolutional Neural Network (CNN) that consists of three convolutional layers and two fully connected (FC) layers without any details of its architecture, purpose, training, etc. It is not clear about the size of the layers as of the number of neurons. The authors did not mention what kind of kernels they used in the convolutional layers, and what features they tried to extract using the convolutions.

Answer: Thank you to the reviewer for your valuable comments and suggestions. It was our oversight not to provide a detailed explanation of the CNN.

We have added a dedicated section (Lines 165–212) to address and strengthen this aspect. The section includes:

(1) Model Architecture

(2) Dataset and Labeling

(3) Training, Validation, and Testing

(4) Performance Evaluation

This section also clarifies the purpose of using convolutions: primarily, to extract the final features of the eyelash extensions for assessing durability.

POSITION: Line 165-212

  1. The authors claimed 93.2% accuracy on the validation set but did not say anything about the accuracy achieved on the testing set which is the required set for accuracy assessment. A testing set is needed to assess the accuracy. According to the authors, their dataset consisted of 1,200 labeled samples in the preliminary test. What was meant by “preliminary” tests?

Answer: In the revised manuscript, we have provided a detailed explanation of how accuracy assessment was conducted (Lines 206–212), along with a description of how precision testing was achieved (Lines 494–510).

The "preliminary test" in the first submitted version of the manuscript was a typographical error; we have removed it.

POSITION: Line 206-212; Line 494-510

  1. It was also unclear what kind of accuracy assessment methodology was used. A question also arises whether the dataset was balanced or not.

Answer: Thank you to the reviewer for the helpful reminder. Following your suggestion, we have explained the accuracy assessment methodology and addressed whether the dataset is balanced. POSITION: Line 206-212

  1. It looks that AI was mentioned in the title as a buzzword to promote the paper rather than to describe the actual content of the paper.

Answer: 4. Following your suggestions, we have made the following improvements:

(1) revised the manuscript title to "AIoT-Based Eyelash Extension Durability Evaluation Using LabVIEW Data Analysis" (Lines 2–3) to fit the content presented in the paper.

(2) added a new section detailing CNN-related content in the revised version (Lines 165–212).

(3) included the results obtained through CNN in the Research Results and Discussion section (Lines 494–510).

(4) added a statement in the Conclusion section highlighting the value of the applied approach (Lines 823–828).

  1. Figure 1 (line 247) describes the conceptual framework of the research. However, even in this paper the AI block is shown without details and connected to the rest of the framework by a “blind” dotted line that makes it kind of weakly related to the framework. The diagram shown in Figure 1 is hardly understandable because it is not clear how the information flows between the components – some block in the diagram are shown as dead-end nodes. The figure should be revised to make it more informative.

Answer:  In response to your comments and suggestions, we have made the following revisions:

(1) We have removed Figure 1 from the originally submitted manuscript, as it was highly similar to Figure 3. It has been replaced with: “Figure 1. CNN Experiment Flow Chart” (Lines 170–177).

(2) Regarding the reviewer’s concern about the lack of signal flow in the system architecture (Figure 3) of the AIoT experimental device platform, we have added the information flow between the main functional components. This revision ensures that no dead-end situations remain. Please refer to the updated manuscript (Lines 303–305) for details.

  1. The AI part of the paper is just briefly discussed in the conclusion (lines 867-871) like it was a sideline unimportant device used in the research.

Answer: Thank you for the helpful reminder; it was our oversight. We have made the following improvements:

(1) Revised the manuscript title (Lines 2–3) to better reflect the content presented in the paper.

(2) In the revised version, we added a section explaining the theoretical foundation of CNN (Lines 165–212) and the results obtained through CNN application (Lines 522–539).

  1. The paper is structured with three levels of headings. In my opinion it is too much (deep) for a paper of such size. However, it is subject to personal taste and publisher’s formatting rules.

Answer: The revised manuscript has been adjusted to align with the journal's standard format. Please refer to the attached revised version.

  1. My recommendations are as follows:

1) If the authors want to keep this paper as a AI-related paper, they have to add more information about the AI part of the paper, i.e. neural network architecture, inputs, training, validation, and testing datasets and their proportional sizes. The explanation of whether the dataset was balanced, and the accuracy assessment methodology should be added too.

Answer: Regarding the missing elements in the originally submitted manuscript, we have followed your suggestions to strengthen the AI-related sections in three key areas:

(1) Enhanced the theoretical foundation (Lines 165–212)

(2) Added the results obtained through the CNN application (Lines 522–539)

(3) Included a discussion of its value in the research conclusion (Lines 856–861)

  1. My recommendations are as follows:

2) If the authors are not intended to focus on the AI part of the paper, they should change the paper title to correctly reflect the content of the paper.

Answer: Thank you to the reviewer for your guidance. Per the revised manuscript, we have made more comprehensive revisions based on your comments and suggestions.

  1. The paper needs revision.

Answer: Thank you to the reviewer for your comments. Following your suggestions, we have made thorough revisions and additions throughout the manuscript. The changes have been highlighted in red for your review.

Reviewer 2 Report

Comments and Suggestions for Authors

Dear Authors,
This paper offers valuable contributions to the fields of cosmetology and AIoT (Artificial Intelligence of Things) applications, particularly in the context of assessing eyelash extension durability. The primary strength of the work lies in the development and validation of the AIoTEDP, an innovative AIoT-integrated system designed to objectively and quantitatively evaluate the durability of eyelash extensions under various environmental conditions.
Following a comprehensive review of the manuscript, I would like to provide the following comments for your consideration:
General Comments:
Manuscript Structure: The manuscript contains several subsections. Authors are advised to review all sections and subsections to ensure the structure is coherent, follows a logical sequence, and aligns with the journal's organizational guidelines. The suggested format includes Abstract, Keywords, Introduction, Materials and Methods, Results, Discussion, and Conclusions (optional).
Specific Comments:
Title: The existing title, "AIoT Experimental Device and LabVIEW Analysis - A Study on Its Application in Evaluating the Durability of Eyelash Extensions," is rather lengthy. I recommend revising it to achieve greater conciseness and impact. Additionally, please note a potential formatting issue regarding a "running title." The journal's author guidelines advise against using running titles; kindly refer to: https://www.mdpi.com/authors/layout for further instructions on manuscript layout and style.
Introduction: The Introduction would benefit from a more thorough articulation of the research problem. Please include additional details concerning the specific challenges associated with evaluating eyelash extension durability. It is also important to clearly justify the necessity of developing the Artificial Intelligence of Things Experimental Device (AIoTEDP), highlighting its relevance to the present study and its significance for the field of cosmetology. Furthermore, please refrain from dividing the introduction into subsections as per journal requirements.
Literature Review: The "Literature Review" section presents three research areas but does not provide justification for their selection or explain their relevance to the current study. It is recommended to clearly describe the methodology used to select these works and to state the relationship between each cited study and the paper. Including more recent information about the state of the art may strengthen the foundation of the research and reflect a comprehensive understanding of the scientific literature.
Figures and Tables: Kindly review all figures and tables to ensure they are appropriately sized and conform to the manuscript’s margins. Please consult the journal's "Information for Authors" for detailed guidance on formatting images and tables.
Addressing these recommendations will enhance both the quality of the manuscript and its alignment with the journal’s standards.

Sincerely,

Author Response

Sensors REVIEWER 2

  1. This paper offers valuable contributions to the fields of cosmetology and AIoT (Artificial Intelligence of Things) applications, particularly in the context of assessing eyelash extension durability. The primary strength of the work lies in the development and validation of the AIoTEDP, an innovative AIoT-integrated system designed to objectively and quantitatively evaluate the durability of eyelash extensions under various environmental conditions.

Following a comprehensive review of the manuscript, I would like to provide the following comments for your consideration:

Answer: Thank you to the reviewer for your support and encouragement of this paper. We will make thorough revisions based on your suggestions, and we hope that the revised manuscript meets the journal's requirements.

  1. General Comments:

Manuscript Structure: The manuscript contains several subsections. Authors are advised to review all sections and subsections to ensure the structure is coherent, follows a logical sequence, and aligns with the journal's organizational guidelines. The suggested format includes Abstract, Keywords, Introduction, Materials and Methods, Results, Discussion, and Conclusions (optional).

Answer: We have revised the manuscript according to your recommendations to meet the journal’s requirements, including updates to the sections: Abstract, Keywords, Introduction, Materials and Methods, Results, Discussion, and Conclusions. Please refer to the attached revised manuscript for details.

3

  1. Specific Comments:

Title: The existing title, "AIoT Experimen General Comments:

Manuscript Structure: The manuscript contains several subsections. Authors are advised to review all sections and subsections to ensure the structure is coherent, follows a logical sequence, and aligns with the journal's organizational guidelines. The suggested format includes Abstract, Keywords, Introduction, Materials and Methods, Results, Discussion, and Conclusions (optional).

Answer: Thank you for your valuable comments and suggestions. As noted above, we have revised the manuscript structure according to your recommendations, aligning it with the key sections required by the journal, including: Abstract, Keywords, Introduction, Materials and Methods, Results, Discussion, and Conclusions. Please refer to the attached revised manuscript for details.

  1. Specific Comments:

Title: The existing title, "AIoT Experimental Device and LabVIEW Analysis - A Study on Its Application in Evaluating the Durability of Eyelash Extensions," is rather lengthy. I recommend revising it to achieve greater conciseness and impact. Additionally, please note a potential formatting issue regarding a "running title." The journal's author guidelines advise against using running titles; kindly refer to: https://www.mdpi.com/authors/layout for further instructions on manuscript layout and style.

Answer: We have shortened the manuscript title for greater precision: "AIoT-Based Eyelash Extension Durability Evaluation Using LabVIEW Data Analysis" (Lines 2–3).

  1. Introduction: The Introduction would benefit from a more thorough articulation of the research problem. Please include additional details concerning the specific challenges associated with evaluating eyelash extension durability. It is also important to clearly justify the necessity of developing the Artificial Intelligence of Things Experimental Device (AIoTEDP), highlighting its relevance to the present study and its significance for the field of cosmetology. Furthermore, please refrain from dividing the introduction into subsections as per journal requirements.

Answer: We have further condensed and integrated the Introduction section, incorporating the research objectives into the introductory paragraphs (Lines 56–62) to enhance conciseness and better align with the journal’s requirements.

  1. Literature Review : The "Literature Review" section presents three research areas but does not provide justification for their selection or explain their relevance to the current study. It is recommended to clearly describe the methodology used to select these works and to state the relationship between each cited study and the paper. Including more recent information about the state of the art may strengthen the foundation of the research and reflect a comprehensive understanding of the scientific literature.

Answer: We revised the Literature Review to reflect the key research variables emphasized in the newly updated manuscript title, strengthening the theoretical foundation (Lines 202–236).

Additionally, in the Methodology section, we have incorporated a discussion on the theoretical basis of CNN, as recommended (Lines 165–212).

  1. Figures and Tables: Kindly review all figures and tables to ensure they are appropriately sized and conform to the manuscript’s margins. Please consult the journal's "Information for Authors" for detailed guidance on formatting images and tables.

Addressing these recommendations will enhance both the quality of the manuscript and its alignment with the journal’s standards.

Answer: Thank you for your comments and suggestions. In the revised manuscript, we have made the necessary corrections to the formatting and dimensions of the Figures and Tables (Lines 331–333; 503–506; 554–557; 680–683).

Round 2

Reviewer 1 Report

Comments and Suggestions for Authors

Lines 173-175:  Sentence “The CNN experimental process of this study is divided 173 into five steps [12-14], as shown in Figure 1: (1) Load Dataset; (2) Define the CNN Model; 174 (3) Fit the CNN Model; (4) Preset the CNN Model; and (5) Evaluate the CNN model” is not needed at all because it is to a degree confusing. Please remove it. Also, Figure 1 should be removed for the same reason.

Line179: Replace “(CNN)”-based with just “CNN”

Lines 187-190: It would be nice if the authors provide some clarification on the purpose of the convolutional layers and the features they want to extract with the used kernels.

Lines 200-204: The dataset was randomly divided into the training, validation, and testing sets. The authors wrote that their dataset was balanced by 400 images in each category (lines 195-197). It would be helpful to mention whether the random selection of the training, validation, and testing sets preserved the balance, i.e. kept the training, validation, and testing sets balanced.

Author Response

Second Review Comments

Review 1

1、Lines 173-175:  Sentence “The CNN experimental process of this study is divided 173 into five steps [12-14], as shown in Figure 1: (1) Load Dataset; (2) Define the CNN Model; 174 (3) Fit the CNN Model; (4) Preset the CNN Model; and (5) Evaluate the CNN model” is not needed at all because it is to a degree confusing. Please remove it. Also, Figure 1 should be removed for the same reason.

Answer: In accordance with the reviewer’s suggestions, the above text and Figure 1 have been deleted, and the subsection (lines 164–174) has been optimized.

2、Line179: Replace “(CNN)”-based with just “CNN”

Answer: Revisions have been made according to the reviewer’s request.

3、Lines 187-190: It would be nice if the authors provide some clarification on the purpose of the convolutional layers and the features they want to extract with the used kernels.

Answer: In accordance with the reviewer’s suggestion, CNN-based feature extraction has been added and marked in red font (lines 184–192).

4、Lines 200-204: The dataset was randomly divided into the training, validation, and testing sets. The authors wrote that their dataset was balanced by 400 images in each category (lines 195-197). It would be helpful to mention whether the random selection of the training, validation, and testing sets preserved the balance, i.e. kept the training, validation, and testing sets balanced.

Answer: In accordance with the reviewer’s suggestion, explanations of the CNN training set, validation set, and test set have been added and marked in red font (lines 198–202).

Reviewer 2 Report

Comments and Suggestions for Authors

Dear Authors,

Thank you for your continued efforts on the manuscript titled "AIoT-Based Eyelash Extension Durability Evaluation Using LabVIEW Data Analysis". Upon reviewing the new version, it appears that several key points from my initial review have not been fully addressed. It is crucial to address these suggestions to enhance the overall quality of your work.

The structure of the manuscript remains problematic due to the presence of unnecessary subsections, which affects focus and readability. I strongly advise revisiting the organization to achieve a more cohesive and structured narrative. Additionally, the formatting issues with figures and tables continue to be a concern, and it is imperative that these elements conform to the journal's guidelines. The literature review section also needs further attention; it still lacks a clear presentation of the criteria for selecting works, which is essential to demonstrate the relevance and contribution of your research to the field. Moreover, formatting inconsistencies in the references and sections following the conclusion need to be rectified to meet professional standards. I highly recommend revisiting the manuscript to address all these areas comprehensively, thereby elevating it to meet publication requirements. Your attention to these matters is greatly appreciated, and I look forward to your next submission, reflecting these improvements.

Sincerely,

Author Response

Second Review Comments

Review 2

Thank you for your continued efforts on the manuscript titled "AIoT-Based Eyelash Extension Durability Evaluation Using LabVIEW Data Analysis". Upon reviewing the new version, it appears that several key points from my initial review have not been fully addressed. It is crucial to address these suggestions to enhance the overall quality of your work.

The structure of the manuscript remains problematic due to the presence of unnecessary subsections, which affects focus and readability. I strongly advise revisiting the organization to achieve a more cohesive and structured narrative. Additionally, the formatting issues with figures and tables continue to be a concern, and it is imperative that these elements conform to the journal's guidelines. The literature review section also needs further attention; it still lacks a clear presentation of the criteria for selecting works, which is essential to demonstrate the relevance and contribution of your research to the field. Moreover, formatting inconsistencies in the references and sections following the conclusion need to be rectified to meet professional standards. I highly recommend revisiting the manuscript to address all these areas comprehensively, thereby elevating it to meet publication requirements. Your attention to these matters is greatly appreciated, and I look forward to your next submission, reflecting these improvements.

Supplementary Review 2 First Review Requirements

Sensors REVIEWER 2

1、This paper offers valuable contributions to the fields of cosmetology and AIoT (Artificial Intelligence of Things) applications, particularly in the context of assessing eyelash extension durability. The primary strength of the work lies in the development and validation of the AIoTEDP, an innovative AIoT-integrated system designed to objectively and quantitatively evaluate the durability of eyelash extensions under various environmental conditions.

Following a comprehensive review of the manuscript, I would like to provide the following comments for your consideration:

2、Manuscript Structure: The manuscript contains several subsections. Authors are advised to review all sections and subsections to ensure the structure is coherent, follows a logical sequence, and aligns with the journal's organizational guidelines. The suggested format includes Abstract, Keywords, Introduction, Materials and Methods, Results, Discussion, and Conclusions (optional).

3、(1) Title: The existing title, "AIoT Experiment General Comments:

Manuscript Structure: The manuscript contains several subsections. Authors are advised to review all sections and subsections to ensure the structure is coherent, follows a logical sequence, and aligns with the journal's organizational guidelines. The suggested format includes Abstract, Keywords, Introduction, Materials and Methods, Results, Discussion, and Conclusions (optional).

(2) Title: The existing title, "AIoT Experimental Device and LabVIEW Analysis - A Study on Its Application in Evaluating the Durability of Eyelash Extensions," is rather lengthy. I recommend revising it to achieve greater conciseness and impact. Additionally, please note a potential formatting issue regarding a "running title." The journal's author guidelines advise against using running titles; kindly refer to: https://www.mdpi.com/authors/layout for further instructions on manuscript layout and style.

4、Introduction: The Introduction would benefit from a more thorough articulation of the research problem. Please include additional details concerning the specific challenges associated with evaluating eyelash extension durability. It is also important to clearly justify the necessity of developing the Artificial Intelligence of Things Experimental Device (AIoTEDP), highlighting its relevance to the present study and its significance for the field of cosmetology. Furthermore, please refrain from dividing the introduction into subsections as per journal requirements.

5、Literature Review : The "Literature Review" section presents three research areas but does not provide justification for their selection or explain their relevance to the current study. It is recommended to clearly describe the methodology used to select these works and to state the relationship between each cited study and the paper. Including more recent information about the state of the art may strengthen the foundation of the research and reflect a comprehensive understanding of the scientific literature.

6、Figures and Tables: Kindly review all figures and tables to ensure they are appropriately sized and conform to the manuscript’s margins. Please consult the journal's "Information for Authors" for detailed guidance on formatting images and tables.

Addressing these recommendations will enhance both the quality of the manuscript and its alignment with the journal’s standards.

  1. The structure of the manuscript remains problematic due to the presence of unnecessary subsections, which affects focus and readability

Answer: In accordance with the standard MDPI structure—Abstract → Keywords → Introduction → Materials and Methods → Results → Discussion → Conclusions—we have made the following integrations and optimizations:
(1) The original section 1.1 “Research Motivation and Objectives” has been merged into the Introduction to avoid excessive subdivision.
(2) Sections 3.2.1 and 3.2.2 have been streamlined and integrated;
(3) Subsections within section 4.3.4 have been optimized and consolidated;
(4) For other subsections, the authors believe that their current categorization is necessary for the clarity of the paper’s discussion, and respectfully request to retain the existing structure.

  1. The literature review section also needs further attention; it still lacks a clear presentation of the criteria for selecting works, which is essential to demonstrate the relevance and contribution of your research to the field.

Answer: Regarding the relevance between the literature and the research contributions, we have established coherence throughout the manuscript to highlight the contributions of this study. Additionally, we have made corresponding additions in the Literature Review section, which are marked in red font (lines 133–136).

  1. The formatting issues with figures and tables continue to be a concern, and it is imperative that these elements conform to the journal's guidelines.

Answer: In accordance with the reference guidelines provided by the reviewer (MDPI Layout Style Guide), we have adjusted and revised the figures and tables to comply with the required formatting standards.( https://www.mdpi.com/authors/layout ).

  1. Formatting inconsistencies in the references and sections following the conclusion need to be rectified to meet professional standards.

Answer: The citation format has been standardized to comply with MDPI guidelines, and corresponding adjustments and revisions have been made.